# Additive Interactions between Betulinic Acid and Two Taxanes in In Vitro Tests against Four Human Malignant Melanoma Cell Lines

**DOI:** 10.3390/ijms23179641

**Published:** 2022-08-25

**Authors:** Paula Wróblewska-Łuczka, Justyna Cabaj, Weronika Bąk, Julia Bargieł, Aneta Grabarska, Agnieszka Góralczyk, Jarogniew J. Łuszczki

**Affiliations:** 1Department of Pathophysiology, Medical University of Lublin, 20-090 Lublin, Poland; 2Department of Biochemistry and Molecular Biology, Medical University of Lublin, 20-090 Lublin, Poland

**Keywords:** melanoma, betulinic acid, paclitaxel, docetaxel, drug interaction

## Abstract

The incidence of melanoma is steadily increasing worldwide. Melanoma is the most lethal skin cancer, and new therapeutic methods are being sought. Our research aimed to investigate the cytotoxic and antiproliferative effects of betulinic acid in vitro, used alone and in combination with taxanes (paclitaxel, docetaxel) in four melanoma cell lines. Isobolographic analysis allowed us to assess the interactions between these compounds. Betulinic acid had no cytotoxic effect on normal human keratinocyte HaCaT cells; the amount of LDH released by them was significantly lower compared to melanoma cell lines. The present study shows that betulinic acid significantly inhibits the growth of melanoma cell lines in vitro. The IC50 values of betulinic acid ranged from 2.21 µM to 15.94 µM against the four melanoma lines. Co-treatment of betulinic acid with paclitaxel or docetaxel generated desirable drug–drug interactions, such as an additive and additive with a tendency to synergy interactions.

## 1. Introduction

Melanoma is a malignant neoplasm that develops from neuroectodermal melanocytic cells. Since melanocytes are found in different places in the body, melanoma can occur in many places. The skin is of course the most common, but it can also be found in the digestive tract, genitals, urinary tract, mucous membranes, meninges, and the uveal lining of the eye. The main risk factor for the development of melanoma is exposure to UVB radiation [1]. People with fair complexion, light-colored eyes, and red or blonde hair are particularly vulnerable [2,3]. In addition, risk factors include multiple birthmarks on the skin, male gender, and age over 40. The environmental factor is exposure to ultraviolet radiation, which is associated with a greater number of cases among Caucasian people [3,4,5]. Genetic predisposition also plays an important role in the pathogenesis of melanoma. The most common mutation in patients with hereditary melanoma is the CDKN2 mutation. Others are, for example, mutations in the CDK4 and TP53 genes. Neoplastic transformation as a result of mutations takes place in melanocytes. BRAF V600 mutation is frequent because it is detected in about 50% of cases. It consists in replacing amino acids, most often from valine to glutamic acid (V600E), which results in an increase in kinase activity and ultimately leads to the uncontrolled proliferation of melanocytes [6]. The NRAS mutation is present in 15–20% of cases. Metastatic melanoma progression is often associated with mutations in the PTEN or CCND1 gene [7]. Melanomas whose cells have mutations in the TERT gene are associated with poor prognosis [8].

The incidence of melanoma is steadily increasing worldwide [9]. Melanoma is the deadliest of skin cancers. This is due to the high metastatic potential and the limited response to chemotherapy and radiotherapy. The early stages of melanoma can be very effectively excised surgically, but when the disease is advanced enough for the patient to have distant metastases, the 5-year survival rate drops dramatically to 1–2%; the median survival time is only 6–9 months [10].

Currently, the treatment of patients, in addition to surgical methods, is based on the methods of chemotherapy and immunotherapy. In recent years, new drugs have been introduced to treat melanoma, but scientists are still looking for new treatment options to improve therapeutic outcomes and increase patient survival.

One group of drugs that are tested for activity against melanoma cells is taxanes. Taxanes are very effective cytostatics belonging to the group of mitotic poisons. The antitumor activity of these drugs was discovered in 1978 and for many years they have been the basis for the treatment of many types of cancer, such as breast, ovarian, prostate, stomach, head and neck, and non-small cell lung cancer [11]. The group of taxanes includes the oldest and most widely used—paclitaxel and its stronger, more water-soluble derivative—docetaxel. The first one is derived from the *Taxus bravifolia* short-leaved yew extract, and the second is derived from a substance present in the common yew. Taxanes belong to the group of terpenes [12].

Taxanes are already being used successfully in the treatment of melanoma [13,14,15]. Their use is most often associated with a combination with other drugs in order to improve therapeutic effects while reducing side effects [16,17]. Therefore, it was decided to combine the taxanes with the betulin derivative due to its numerous pharmacological effects. Betulin is a compound belonging to the group of triterpenes, a component of birch bark extract. Betulin oxidation product—betulinic acid—is now a readily researched compound because it has many desirable properties. By influencing proteins from the Bcl-2 family and increasing the level of reactive oxygen species, betulinic acid induces apoptosis, which is often inhibited in neoplastic cells. Due to this, abnormal cells will undergo the process of programmed death, and will not divide further, passing on the defects to daughter cells [18]. Some cancer cells are insensitive to the therapeutic effects of drugs or radiotherapy. Betulinic acid has been shown to be able to sensitize cancer cells to chemotherapy or radiation, which were previously resistant to them. The experiments showed that betulinic acid increased the radiation sensitivity of oral squamous cell carcinoma cavity through Sp1 sumoylation. This increased the expression of PTEN protein, which is an important tumor suppressor gene [19]. Another experiment showed that betulinic acid facilitated the process of apoptosis in paclitaxel-resistant lung cancer cells [20]. Other reports show that combining 5-fluorouracil with betulinic acid stops hedgehog signaling, which has a strong influence on chemoresistance. The combination of these two substances increased chemosensitivity by inhibiting the GLI1, GLI2, and PTCH1 genes in ovarian cancer cells [21]. In addition, betulinic acid reduces angiogenesis—a process that promotes tumor growth and metastasis and induces autophagy [22]. Our study aimed to investigate cytotoxic and anti-proliferative effects of betulinic acid (BA) in vitro when used alone and when combined with taxanes (paclitaxel, docetaxel) in four melanoma cell lines. The detailed pharmacodynamic interactions between combinations of BA with paclitaxel and BA with docetaxel were analyzed by the isobolographic analysis.

## 2. Results

### 2.1. Cell Viability of BA and Taxanes on Malignant Melanoma Cancer Cell Lines

Cell viability (e.g., MTT assay) studies are typically the first step in determining a potential anticancer activity of new compounds. The effect of BA, paclitaxel, and docetaxel on the viability of cells was assessed in the MTT test, based on mitochondrial enzyme activity.

In the present study, four melanoma cell lines (A375, SK-MEL28, FM55P, and FM55M2) and normal human keratinocytes (HaCaT) were treated with various concentrations of BA. As shown in Figure 1, BA reduced the viability of the studied melanoma cells in a concentration-dependent manner.

Statistically significant inhibition of the viability of A375, SK-MEL28, and FM55M2 was observed in the concentration range of 1–40 µM of BA, and for FM55P cells in the concentration range of 2–40 µM of BA. In the case of normal human keratinocytes (HaCaT), significant, albeit slight inhibition of cell viability was observed at high BA concentrations of 16–40 µM.

Both, paclitaxel (Figure 2) and docetaxel (Figure 3) showed decreased viability of the studied melanoma cells in a concentration-dependent manner. In contrast to the rest of the melanoma cell lines, A375 cells were relatively more resistant to the treatment with taxanes. An inhibition of cell proliferation was observed at higher concentrations compared to other melanoma cell lines. The final concentration of dimethyl sulfoxide (DMSO) in the culture medium used to dissolve the betulinic acid, paclitaxel, and docetaxel did not exceed 0.1% and did not affect cell viability and cell membrane integrity (data not shown).

Paclitaxel and docetaxel are registered drugs used in clinical practice. The concentrations tested by us for paclitaxel 0.001–0.5 µM and for docetaxel 0.0005–0.08 µM are significantly lower than the maximum plasma concentrations (C_max_) in clinical practice, which are 4.27 µM and 5.47 µM for paclitaxel and docetaxel, respectively [23].

### 2.2. Cytotoxicity of Betulinic Acid—LDH Test Result

Cytotoxicity of betulinic acid (BA) to normal human keratinocytes (HaCaT) and malignant melanoma cells (A375, SK-MEL28, FM55P, and FM55M2) was measured by LDH assay. The LDH test allows us to assess the release of lactate dehydrogenase into the medium, which indicates damage to the cell membrane and cell death [24]. In our experiment, a significant leakage of LDH was observed in cell lines A375 and SK-MEL28 treated with BA in the concentration range of 16–40 µM. An important issue is that betulinic acid did not exert a cytotoxic effect on normal human keratinocyte cells HaCaT, the amount of LDH released by them was significantly lower compared to the melanoma cell lines (Figure 4).

### 2.3. Flow Cytometry Analysis

Apoptosis is considered to be one of the main mechanisms of anti-cancer defense [25]. In order to determine whether the anti-proliferative effect of betulinic acid (BA) in melanoma cells was associated with apoptosis induction, we have measured a population of cells with activated caspase 3 by flow cytometry. Caspase-3 is a well-described protease and its activation is a hallmark of apoptosis [26]. In our studies, we have found an increased number of caspase-3 positive cells versus control after treatment with betulinic acid at its IC_50_ concentration (Figure 5). The cytotoxicity of betulinic acid-inducing apoptosis may be related to the massive accumulation of reactive oxygen species (ROS) in cells. BA-mediated overproduction of ROS plays a key role in the anti-tumor activity of this compound, which is molecularly associated with significant inhibition of the inappropriately activated NF-κB pathway [27,28].

### 2.4. Isobolographic Analysis of Interaction between BA and Taxanes in Melanoma Cell Lines

Next, the median inhibitory concentrations (IC_50_ values) were calculated from equations of log-probit concentration–response relationship curves (CRRCs) for BA and Paclitaxel (Figure 6A–D) and for BA and Docetaxel (Figure 7A–D). They are summarized in Table 1. We observed that BA, Paclitaxel, and Docetaxel showed an antiproliferative effect on the melanoma cells, with IC_50_ values ranging from 2.21 ± 0.42 to 15.94 ± 3.95 µM, from 4.63 ± 0.62 to 96.20 ± 14.61 µM, and from 1.27 ± 0.55 to 15.83 ± 9.05 µM, respectively.

The test for parallelism between BA and Paclitaxel revealed that the lines of both compounds were non-parallel to each other in the A375, SK-MEL28, and FM55M2 cell lines and parallel to one another in the FM55P (Figure 6). In turn, the test for parallelism between BA and Docetaxel revealed that the lines of both compounds were parallel to each other only in the A375 cell line (Figure 7).

Isobolographic analysis of interactions between BA and paclitaxel revealed that in vitro combinations were additive in every tested melanoma cell line (Figure 8A–D). In the case of a combination of BA and docetaxel (Figure 9A–D), isobolograms showed an addition with a tendency to synergy for A375 cell line and additive interactions for the remaining tested melanoma cell lines (FM55P, FM55M2, and SK-MEL28).

## 3. Discussion

Melanoma is a cancer that is diagnosed easier and faster over time. The growing number of cases means that new methods of treatment are sought, especially in the case of severe course, metastases, or resistance to therapy [10]. Compounds of natural origin with anti-cancer properties are sought. Terpene compounds are tested, such as betulin and its derivatives, alkaloids [29], and coumarins, which show the effectiveness of melanoma therapy [30].

Betulinic acid (BA) and its derivatives have a number of pharmacological properties (including, antidiabetic, anti dislipidemic, anxiolytic, antidepressant, anti-inflammatory, antiviral, antibacterial, and anticancer effects). BA is considered to be a safe compound with low toxicity when tested both, in in vitro and in vivo models. Evidence shows that the toxic effects of BA on cancer cells are greater than its effects on normal cells, which is also confirmed by the results of our research. Moreover, the in vivo attainable concentration of BA has the ability to arrest colony formation by tumor cells. Interestingly, BA showed no serious adverse effects such as systemic toxicity and weight loss when administered to animals at high doses [31].

To date, several targets and molecular mechanisms of BA anti-tumor activity have been proposed. However, the exact molecular mechanism of BA remains unknown, and researchers believe that BA exerts its effects through different pathways such as induction of apoptosis, regulation of autophagy machinery, inhibition of angiogenesis and metastasis, and chemo-sensitization mechanisms [22].

Betulinic acid has been shown to selectively affect a wide range of tumor cell lines both, in vivo and in vitro. Additionally, many studies have shown that BA has no cytotoxic effects on normal cells such as fibroblasts, peripheral blood lymphoblasts, melanocytes, and astrocytes [31]. The LDH test performed herein confirmed that BA was not cytotoxic on keratinocytes. The anti-tumor effects of BA were initially observed on human melanoma cells with a reported IC_50_ = 1.5–1.6 µg/mL (Me665/2/21 and Me665/2/60 melanoma cell lines) [31,32]. Then its effects were reported on other tumor cell lines including neuroblastoma (IC_50_ = 14–17 µg/mL), medulloblastoma (IC_50_ = 3–13.5 µg/mL), Ewing’s sarcoma, leukemia, brain tumors, glioma (IC_50_ = 2–17 µg/mL), colon cancer, ovarian cancer (IC_50_ = 1.8–4.5 µg/mL), lung cancer (IC_50_ = 1.5–4.2 µg/mL), breast cancer, prostate cancer, hepatocellular carcinoma, kidney, and cervical cancer (IC_50_ = 1.8 µg/mL) [31,33]. Our experiments showed the concentration-dependent antiproliferative effect of BA on four melanoma cell lines. The IC_50_ values of BA ranged from 2.21 µM to 15.94 µM, which is about 1 µg/mL to 7.3 µg/mL, which is very comparable to the results of other authors. 

The skin is extremely exposed to oxidative stress, which can lead to cancer progression or melanoma chemoresistance. Among the current therapeutic approaches in the treatment of melanoma, there is an increasing interest in compounds that would lead to the accumulation of ROS in cancer cells, including platinum derivatives, brusatol, ailanthone, luteolin, resveratrol, curcumin, and many others [34]. Therefore, such an interesting therapeutic choice seems to be BA, the effectiveness of which in cancer therapy is associated with the ability to accumulate intracellular ROS [27] and participates in the sensitization of cells to chemotherapeutic agents [19,20,21].

One of the described strategies to increase the effectiveness of BA as a promising anti-cancer drug is its combination with other chemotherapeutic agents [35]. Combination therapy involving drugs acting by a different mechanism is currently one of the most promising therapeutic strategies in oncology, which may prevent the occurrence of severe side effects associated with monotherapy and reduce the likelihood of developing cancer cell resistance [36]. Currently, in the treatment of melanoma, combinations of chemotherapeutic agents with coumarins (e.g., osthole) [30], cannabinoids such as CBD [37] or with amantadine (a drug used in Parkinson’s disease) [38] are being tested.

In one study, it was found that the application of the combination of BA with a gamma-cyclodextrin derivative to B164A5 melanoma cells reduces cell proliferation and induces cell death [39]. A combination of 5-fluorouracil (5-FU) and BA has also shown an increased rate of cell apoptosis and morphological changes in the mitochondrial membrane of OVCAR 432 ovarian cancer cells [21].

Another study attempted to test the combination of BA and APO2 on HUN7 and PLC/PRF/5 cells in vitro and in vivo mouse xenograft models, as well as a potential molecular mechanism. APO2 is considered a new therapeutic strategy for the induction of apoptosis in cancer cells and has no apparent effect on normal cells. Nevertheless, research also indicates that many cancer cells are resistant to APO2-induced apoptosis. Combining BA with APO2 has been shown to have the potential value in the fight against liver cancer. Combination treatment with BA and APO2 has been shown to inhibit liver cancer progression in vitro and in vivo by targeting the p53 signaling pathway, increasing apoptosis in APO2-resistant liver cancer cells without toxicity in normal cells or organs [40].

The combination of BA with first-generation tyrosine kinase inhibitors (TKIs) targeted at treating lung cancer patients has shown promising results. More than half of lung cancer patients have mutations in the epidermal growth factor receptor (EGFR). Unfortunately, after several months of therapy, neoplastic cells show resistance to the antitumor effects of gefatinib and erlotinib. The study showed that the combination of BA and TKIs increased cell death in HCC827 and H1975 lung cancer cell lines by increasing the expression of apoptotic proteins and autophagy. Combination therapy with these compounds improved the effectiveness of anti-cancer therapy and reduced the side effects of chemotherapeutic agents [41].

Sorafenib is a multidirectional kinase inhibitor that has therapeutic effects against a variety of cancers by inhibiting VEGFR2-mediated angiogenesis and the RAS/RAF/ERK pathway. Previous studies showed that combination therapy with BA and sorafenib showed a potent and synergistic inhibitory effect on the proliferation of cancer cells such as non-small cell lung cancer (NSCLC), A549, H358, and A427 with different KRAS mutations. Combination therapies with low doses of BA and sorafenib have shown therapeutic potential in the treatment of lung cancer, additionally having no toxic effect on lymphocytes [42,43].

Renal cell carcinoma (RCC), a common type of kidney cancer, is remarkably resistant to radiation therapy, chemotherapy and cytokine immunotherapy (including, interferon α/γ, interleukin-2) with a response rate of less than 20%. These cells are also resistant to 5-FU, temozolomide, and etoposide. Evidence shows that the combination of betulin with 5-FU, temozolomide and etoposide has a significant synergistic effect against drug-resistant RCC cells. The combination of these compounds synergistically inhibited poly(ADP-ribose) polymerase (PARP) expression and increased MDR1 expression in RCC4 cells. Therefore, betulin is a promising candidate for chemopreventive and chemotherapeutic agents in the treatment of multi-drug resistant human renal cancer [44,45].

BA has clearly proven its effectiveness in vitro and in vivo in the treatment of many malignant neoplasms (i.e., melanoma, hepatocellular carcinoma, colorectal, lung, breast, prostate, stomach, pancreas, neck and head, ovary cancers, glioblastoma multiforme, and chronic myeloid leukemia). It has also shown selectivity, with no or very little effect on normal cells even at doses as high as 500 mg/kg body weight [46,47,48]. In addition, BA developed for topical application (20% betulinic acid ointment) was assessed in a phase I/II clinical trial for dysplastic lesions that may evolve into melanoma. Unfortunately, the study was suspended due to financial constraints [49,50].

Taxanes, which are also the subject of the presented research, are very effective cytostatics belonging to the group of mitotic poisons. In addition, it has been proven that, apart from the role of cytostatics, taxanes have many other valuable properties. It has been tested that they have immunomodulatory abilities that increase the effectiveness of chemotherapy. They stimulate macrophages to secrete cytokines, including TNF-α (tumor necrosis factor), interleukins: IL-10 and IL-12, which activate dendritic cells, NK cells, and cytotoxic lymphocytes. They also enhance the cytotoxicity of NK cells through an increased level of perforin [12,17].

It can be seen that combining drugs or substances is still an active trend of searching for new anti-cancer therapies. One of the other authors’ publications also related to a combination of betulinic acid and taxol (synonym: paclitaxel) tested for breast cancer. The study in the MCF-7 and MDA-MB-231 breast cancer cell lines investigated whether BA could synergistically potentiate taxol-induced apoptosis and cell cycle arrest effects. Treatment of breast cancer cells with a low dose of taxol alone induced minimal apoptosis. However, when BA was administered concomitantly with taxol, the rate of apoptosis increased dose-dependently. The synergistic effect of the combination of BA and taxol was confirmed in studies on mice [46]. Paclitaxel-betulinic acid (PTX-BA-NP) hybrid nanosuspensions have been developed with increased anti-cancer activity, which has been proven in breast cancer research (MCF-7 cell line). The combination of these compounds had a better effect than drugs administered separately [51]. The combination of docetaxel and BA showed a beneficial effect on the apoptosis of prostate cancer cells, caused by the increased activity of NF-kappaβ [52]. In melanoma studies, the synergistic effect of the combination of paclitaxel and vinorelbine (G361 and StM111a cell lines) [53] and the antagonistic effect of the combination of paclitaxel and cisplatin (G361 cell line) have been demonstrated [54]. Therefore, it seems so important to determine the interactions between the compounds administered in combination to confirm that, when administered together, they do not abolish their effects. For this purpose, the most useful seems to be the isobolographic analysis, which is a type of statistical analysis referred to as the “gold standard” in determining interactions. When analyzing the publication databases, no studies were found that used isobolographic analysis to assess the interaction of a combination of BA with another anticancer substance. Our study showed for the first time that the mixture of BA with taxanes decreased melanoma cell viability. Moreover, co-treatment of BA with paclitaxel or BA with docetaxel generated desirable drug–drug interaction, such as an additive and additive with a tendency to synergy interactions.

## 4. Materials and Methods

### 4.1. Cell Lines

Primary (FM55P) and metastases (FM55M2) malignant melanoma cells were purchased from the European Collection of Cell Cultures (ECACC, Salisbury, UK) and cultured in RPMI-1640 Medium (Sigma-Aldrich, St. Louis, MO, USA). Another two cell lines A375 (primary malignant melanoma) and SK-MEL 28 (metastatic malignant melanoma) were purchased from the American Type Culture Collection (ATCC, Manassas, Virginia, USA) and cultured in Dulbecco’s Modified Eagle’s Medium—high glucose (DMEM) (Sigma-Aldrich, St. Louis, MO, USA) and Eagle’s minimal essential medium (EMEM), respectively. All culture media were supplemented with 10% Fetal Bovine Serum (FBS; Sigma-Aldrich, USA) and 1% of penicillin/streptomycin (Sigma-Aldrich, St. Louis, MO, USA). Cultures were kept at 37 °C in a humidified atmosphere of 95% air and 5% CO_2_. The cells grew to 80% confluence.

### 4.2. Chemicals

Paclitaxel, docetaxel, and betulinic acid (Sigma-Aldrich, St. Louis, MO, USA) were dissolved in DMSO as stock solutions. The drugs were dissolved to the respective concentrations with a culture medium before use, maintaining the concentration of DMSO up to 0.1%.

### 4.3. Cell Viability Assessment

A375, SK-MEL28, FM55P, and FM55M2 melanoma cells and normal human keratinocytes (HaCaT) cells were placed on 96-well plates (Nunc, Roskilde, Denmark) at a density of 2–3 × 10^4^ cells/mL. The next day, the culture medium was removed and cells were exposed to serial dilutions of paclitaxel, docetaxel, and betulinic acid in a fresh culture medium. Cell viability was assessed after 72 h by means of the MTT method. The 72 h incubation time is the average doubling time for all melanoma cell lines tested. In the case of the A375 line, this time is the shortest 6–12 h [55], for the SK-Mel28 it is 17.5 h [56], but in the case of the FM55M2 and FM55P lines, most of the experiments encountered are the incubation time of 72 h [37,38,57]. After 72 h incubation, cells were incubated for 3 h with MTT solution (5 mg/mL, Sigma-Aldrich, USA). Formazan crystals were solubilized overnight in sodium dodecyl sulfate (SDS) buffer (10% SDS in 0.01 N HCl) and the product was determined spectrophotometrically by measuring absorbance at 570 nm wavelength using a microplate spectrophotometer (Ledetect 96, Labexim, Lengau, Austria). Each experiment was repeated three times.

### 4.4. Cytotoxicity Assessment—LDH Assay

Optimized amounts of four melanoma cell lines (2–3 × 10^4^/mL) and normal human keratinocytes HaCaT (1 × 10^4^/mL) cells were placed on 96-well plates (Nunc, Roskilde, Denmark). The next day, cells were exposed to increasing concentrations of betulinic acid in a fresh culture medium. The cytotoxicity was estimated based on the measurement of cytoplasmic lactate dehydrogenase (LDH) activity released from damaged cells after 72 h exposure to betulinic acid. LDH assay was performed according to the manufacturer’s instruction (Cytotoxicity Detection KitPLUS LDH) (Roche, Mannheim, Germany). Absorbance was measured at two different wavelengths, one being the “measurement wavelength” (492 nm) and the other “reference wavelength” (690 nm) using a microplate spectrophotometer (Ledetect 96, Labexim, Lengau, Austria). Maximum LDH release (positive control) was achieved by the addition of Lysis buffer to untreated control cells. The average values of the culture medium background were subtracted from all values of experimental wells and the percentage of dead cells were calculated in relation to the maximum LDH release.

### 4.5. Apoptosis Analysis

Optimized amounts of malignant melanoma cells were placed on six-well plates (Nunc). After a day, the medium was replaced with a fresh medium containing betulinic acid (BA) at an IC_50_ dose for 72 h. After that, cells were harvested and washed twice with PBS. Next, cells were fixed and permeabilized using the cytofix/cytoperm solution according to the manufacturer’s instructions of Phycoerythrin (PE) Active Caspase-3 Apoptosis Kit (BD Pharmingen). Finally, cells were washed twice in the perm/wash buffer prior to intracellular staining with PE-conjugated anti-active caspase-3 monoclonal rabbit antibodies. Labeled cells were analyzed by flow cytometer FACSCalibur (Becton Dickinson, San Jose, CA, USA), operating with CellQuest software (BD Biosciences, San Jose, California, USA) to quantitatively assess the caspase-3 activity.

### 4.6. Isobolographic Analysis of Interactions

Isobolographic analysis is a statistical method allowing the characterization of pharmacodynamic interactions between drugs and chemical substances. First, the percent inhibition of cell viability per increasing concentrations of betulinic acid (BA), paclitaxel, and docetaxel (when administrated alone in the melanoma cell lines A375, SK-MEL28, FM55P, and FM55M2 cell lines) was measured in vitro by the MTT assay. Then, the concentration-response effects for each investigated anti-cancer compound (i.e., BA, paclitaxel, and docetaxel) were fitted with log-probit linear regression analysis as described by Litchfield and Wilcoxon [58]. Subsequently, from the log-probit concentration-response lines, median inhibitory concentrations (IC_50_ values) of paclitaxel, docetaxel, and BA were calculated. Test for parallelism of two concentration-response lines (BA and paclitaxel, BA, and docetaxel) was performed as described in detail in our previous studies [29,30,59]. The median additive inhibitory concentrations (IC_50 add_) for the mixture of BA with paclitaxel or BA with docetaxel, which theoretically should inhibit 50% of cell viability, were calculated as demonstrated by Tallarida [60,61,62]. The assessment of the experimentally derived IC_50 exp_ at the fixed ratio of 1:1 was based on the concentrations of the mixture of BA and paclitaxel or BA and docetaxel that inhibited 50% of cell viability in melanoma cell lines measured in vitro by the MTT assay. Details concerning the isobolographic analysis have been published elsewhere [29,30,61,63].

### 4.7. Statistical Analysis

GraphPad Prism 8.0 Statistic Software (San Diego, CA, USA) was used for statistical analysis. The calculations were performed by one-way analysis of variance (ANOVA test) for multiple comparisons followed by Tukey’s significance test. Data are expressed as the mean ± standard error of mean (SEM) (* *p* < 0.05, ** *p* < 0.01, *** *p* < 0.001, **** *p* < 0.0001). The IC_50_ and IC_50 exp_ values for BA and paclitaxel or docetaxel, administered alone or in combinations at the fixed ratio of 1:1, were calculated by computer-assisted log-probit analysis [58]. The experimentally derived IC_50 exp_ values for the mixture of BA with paclitaxel and BA with docetaxel were statistically compared with their respective theoretically additive IC_50 add_ values by the use of unpaired Student’s *t*-test [62,63].

## 5. Conclusions

The present study showed that BA significantly inhibited the growth of melanoma cell lines in vitro. By conducting studies on several human melanoma cell lines, we have clearly shown that the anti-tumor activity of BA is not specific to a single cell line and can be generalized to the type of tumor. Moreover, it is the first report indicating the anti-proliferative activity of BA in combination with taxanes against melanoma cells. Additive and additive with a tendency to synergy interaction of BA with paclitaxel and BA with docetaxel, as assessed by isobolographic analysis in human melanoma cell lines. Further intensive research is required to determine the possible mechanism(s) of action of the combination of BA with taxanes and to conduct animal experiments to confirm the interactions of compounds in vivo conditions. Despite BA enormous potential in the treatment and prevention of many diseases, there are few clinical experiments with this compound.

## Figures and Tables

**Figure 1 ijms-23-09641-f001:**
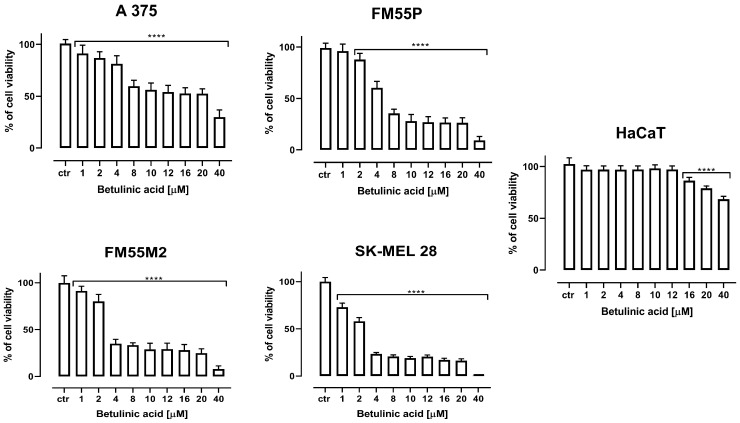
The effect of betulinic acid (BA) on the viability of malignant melanoma cancer cell lines and normal human keratinocytes (HaCaT) cell lines was measured by MTT assay after 72 h. Results are presented as mean ± SEM at each concentration. (**** *p* < 0.0001).

**Figure 2 ijms-23-09641-f002:**
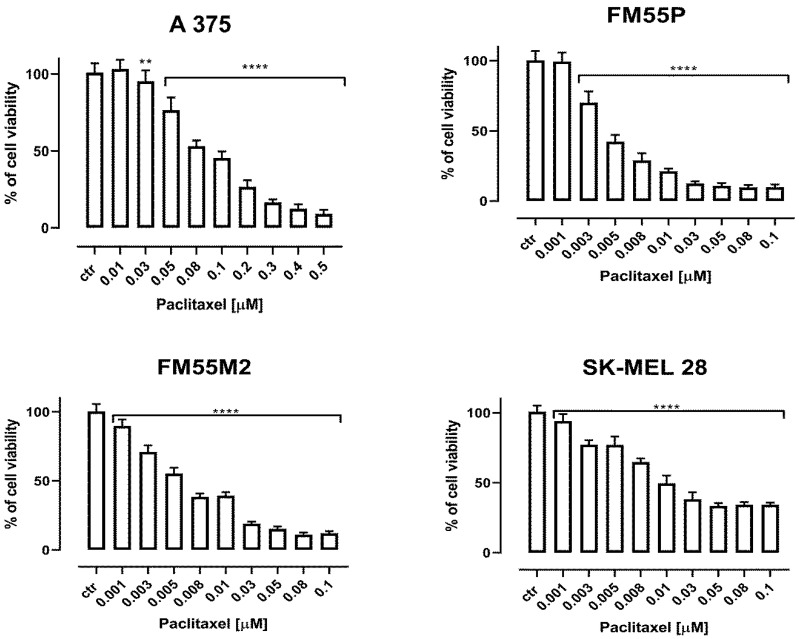
The effect of paclitaxel on the viability of malignant melanoma cancer cell lines was measured by MTT assay after 72 h. Results are presented as mean ± SEM at each concentration. (** *p* < 0.01; **** *p* < 0.0001).

**Figure 3 ijms-23-09641-f003:**
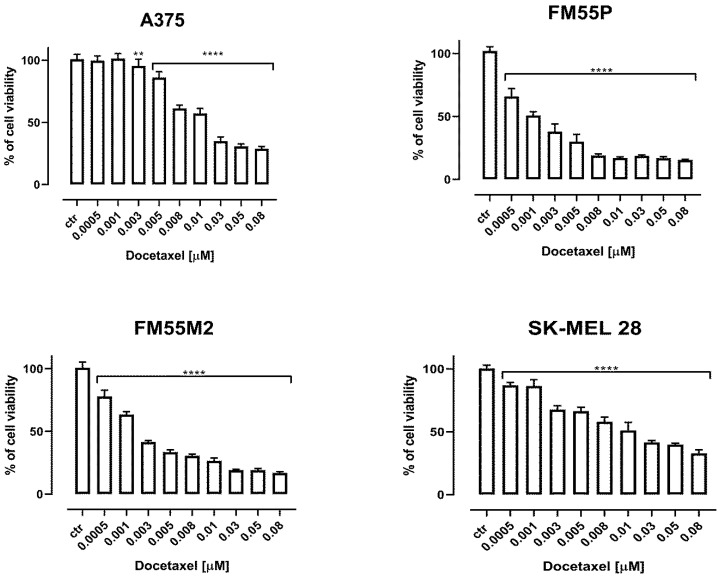
The effect of docetaxel on the viability of malignant melanoma cancer cell lines was measured by MTT assay after 72 h. Results are presented as mean ± SEM at each concentration. (** *p* < 0.01; **** *p* < 0.0001).

**Figure 4 ijms-23-09641-f004:**
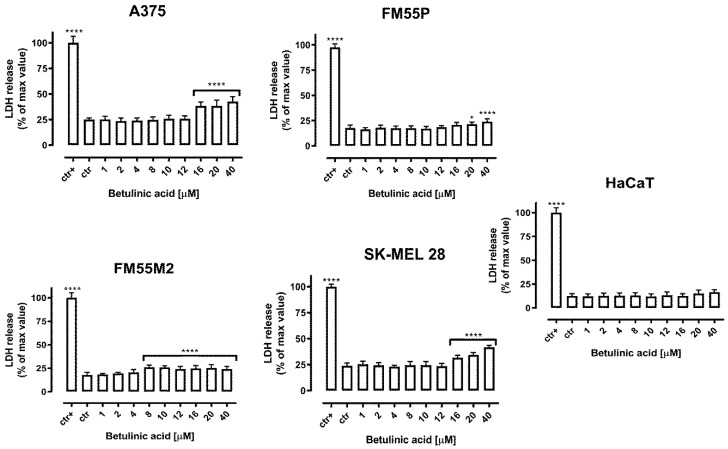
Cytotoxicity of betulinic acid (BA) to melanoma cells (A375, SK-MEL28, FM55P, and FM55M2) and normal human keratinocyte cells (HaCaT). Tested cells were incubated for 72 h alone or in the presence of BA (1–40 µM). The results are presented as the percentage of LDH released to the medium by treated cells vs. cells grown in the control medium (ctr) and cells treated with lysis buffer (ctr+). Results are presented as mean ± SEM at each concentration. (* *p* < 0.05; **** *p* < 0.0001).

**Figure 5 ijms-23-09641-f005:**
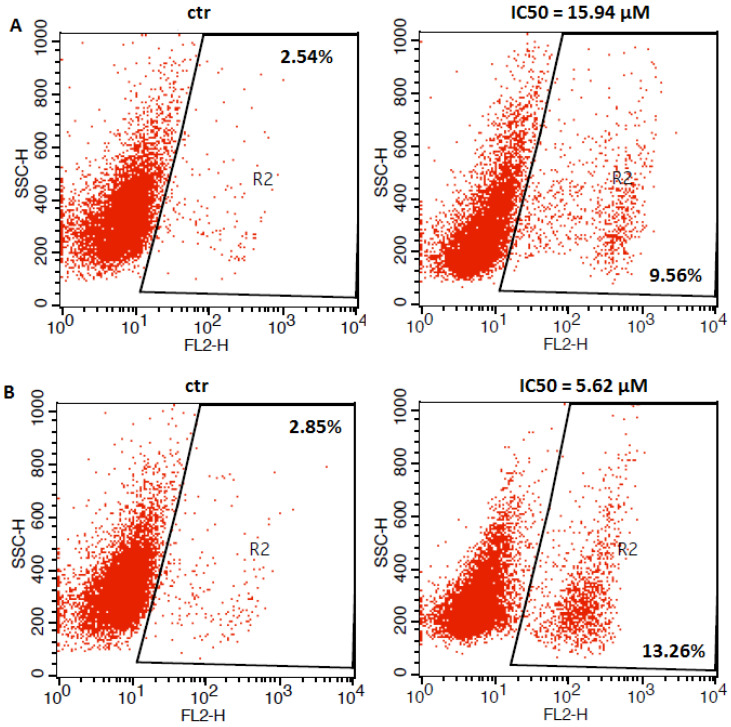
Representative flow cytometry dot plot graphs of A375 (**A**), FM55P (**B**), FM55M2 (**C**), and SK-MEL28 (**D**) melanoma cell lines after the treatment with a medium (ctr) and betulinic acid (BA) at their IC_50_ concentrations for 72 h. Region R2 included apoptotic cells with active caspase-3.

**Figure 6 ijms-23-09641-f006:**
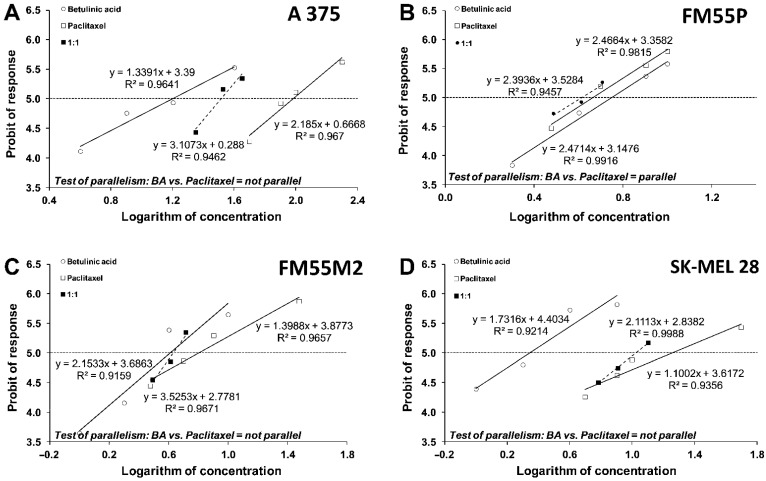
(**A**–**D**). Log-probit lines for betulinic acid (BA) and paclitaxel administered alone, and in combinations at the fixed ratio of 1:1 (dotted line), illustrating the anti-proliferative effects of the drugs in the malignant melanoma cancer cell lines (A375 (**A**), FM55P (**B**), FM55M2 (**C**), and SK-MEL28 (**D**)) measured in vitro by the MTT assay.

**Figure 7 ijms-23-09641-f007:**
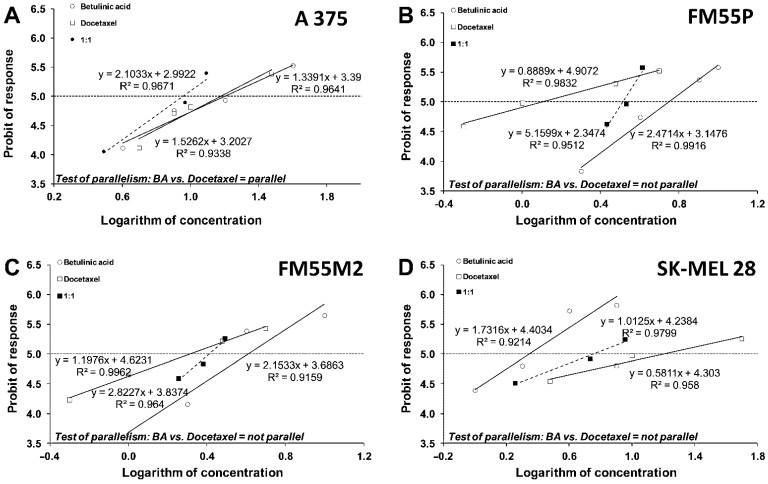
(**A**–**D**). Log-probit dose-response relationship curves (DRRCs) for betulinic acid (BA) and docetaxel administered alone, and in combinations at the fixed ratio of 1:1 (dotted line), illustrating the anti-proliferative effects of the drugs in malignant melanoma cancer cell lines (A375 (**A**), FM55P (**B**), FM55M2 (**C**), and SK-MEL28 (**D**)) measured in vitro by the MTT assay.

**Figure 8 ijms-23-09641-f008:**
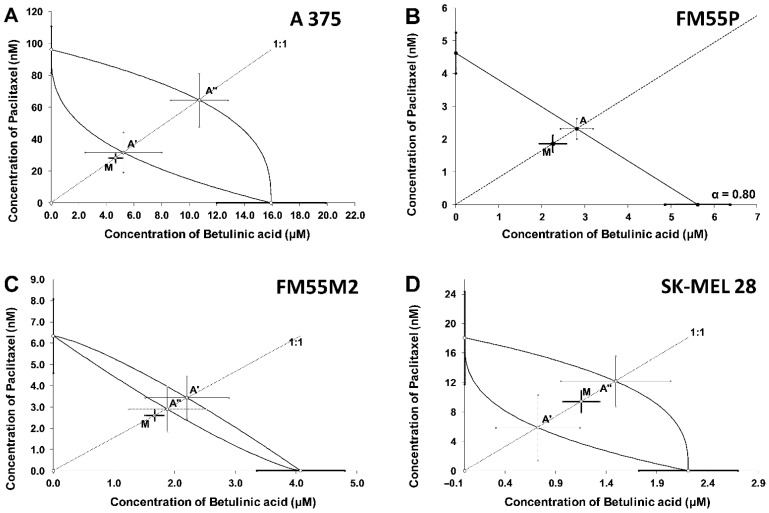
(**A**–**D**). Isobolograms showing additive interactions between betulinic acid (BA) and paclitaxel with respect to their anti-proliferative effects on A375 (**A**), FM55P (**B**), FM55M2 (**C**), and SK-MEL28 (**D**) malignant melanoma cell lines measured in vitro by the MTT assay. The median inhibitory concentrations (IC_50_) for BA and paclitaxel are plotted graphically on the X- and Y-axes, respectively. The lower and upper isoboles of additivity represent the curves connecting the IC_50_ values for BA and paclitaxel administered alone. The dotted line starting from the point (0. 0) corresponds to the fixed ratio of 1:1 for the combination of BA with paclitaxel. The points A, A’ and A” depict the theoretically calculated IC_50 add_ values for both, lower and upper isoboles of additivity. The point M represents the experimentally derived IC_50 exp_ value for a total dose of the mixture expressed as proportions of BA and paclitaxel that produced a 50% anti-proliferative effect (50% isobole) in malignant melanoma cell lines measured in vitro by the MTT assay. On the graph, the S.E.M. values are presented as horizontal and vertical error bars for every IC_50_ value.

**Figure 9 ijms-23-09641-f009:**
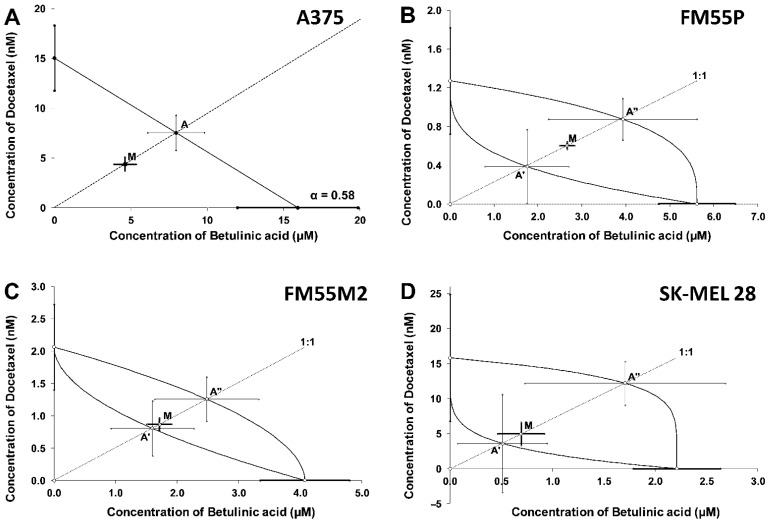
(**A**–**D**). Isobolograms showing interactions between betulinic acid (BA) and docetaxel with respect to their anti-proliferative effects on A375 (**A**), FM55P (**B**), FM55M2 (**C**), and SK-MEL28 (**D**) malignant melanoma cell lines measured in vitro by the MTT assay. The median inhibitory concentrations (IC_50_) for BA and docetaxel are plotted graphically on the X- and Y-axes, respectively. For more details see Figure 6.

**Table 1 ijms-23-09641-t001:** The anti-proliferative effects of betulinic acid (BA), paclitaxel, and docetaxel administered alone in human malignant melanoma cell lines measured in vitro by the MTT assay. Data are median inhibitory concentrations (IC values ± SEM).

Cell Line	Paclitaxel [nM]	Docetaxel [nM]	Betulinic Acid [µM]
A375	96.20 ± 14.61	15.05 ± 3.27	15.94 ± 3.95
FM55P	4.63 ± 0.62	1.27 ± 0.55	5.62 ± 0.87
FM55M2	6.35 ± 1.74	2.06 ± 0.66	4.08 ± 0.72
SK-MEL28	18.06 ± 6.29	15.83 ± 9.05	2.21 ± 0.42

## Data Availability

All data are presented in tabular and graphical forms. Additionally, the data could be available upon request.

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
