# Peer review of "Additive Interactions between Betulinic Acid and Two Taxanes in In Vitro Tests against Four Human Malignant Melanoma Cell Lines"

_ijms, 2022, doi:10.3390/ijms23179641_

Round 1

Reviewer 1 Report

The manuscript “Additive interactions between betulinic acid and two taxanes in in vitro tests against four human malignant melanoma cell lines” is a research article in which authors show that co-treatment of betulinic acid with paclitaxel or with docetaxel induced an additive drug–drug interaction in melanoma cells. Some sentences should be rephrased as they are incorrect or misleading; few typos are present. There are some important concerns that authors must address in order to consider the manuscript suitable for publication:

1.       In the introduction section, more information should be provided about melanoma; for instance, authors correctly state that UV is the main risk factor, but they should also mention male sex, fair skin, amount of moles, and age (PMID: 34638427). Please expand this section.

2.       Line 42: “The incidence of melanoma is steadily increasing worldwide”; this statement requires to be justified by a reference.

3.       Line 61: “Taxanes are already successful in treating melanoma”; this statement requires to be justified by a reference.

4.       Line 61-65: this part is poorly written and sounds not scientific. Please rephrase it.

5.       Line 70: “It was also tested that betulinic acid has the ability to sensitize”; it is not clear to what betulinic acid is able to sensitize cells. Please rephrase and specify.

6.       Line 79 “contributes”.

7.       I do not understand why the measurements were performed at only 72h hours after treatment, and were not monitored over the time at 0, 24, 48 and 72h. Do authors have any data about previous timepoints?

8.       Figure 1: Authors must perform the same experiment on normal human keratinocyte cells HaCaT treated with BA. This is crucial to demonstrate that the reduction of cell viability observed in melanoma cells would not affect normal melanocytes.  

9.       A mechanistic insight would be highly recommended. Is BA administration able to increase intracellular ROS production?

10.   Line 180: “Due to the growing number of cases of melanoma, new therapeutic methods are  sought. It is certainly related to the increased detection of this cancer, as well as the in-181 creasing exposure to ultraviolet (UVB) radiation”. These sentences are not connected each other; please rephrase.

11.   In the discussion section authors should mention also the other therapeutical approaches currently under evaluation for melanoma treatment, including coupling natural compounds and chemotherapeutics (PMID: 35453297). Indeed in line 216 they suggest coupling the BA with chemotherapeutics. This  point will strongly support the author’s design of study.

Author Response

Thank You very much for Your time and Your review of our manuscript.
All comments indicated by the Reviewer were taken into account and the appropriate corrections were made to the manuscript.
We hope You will be satisfied with the corrections we have made.

  1. In the introduction section, more information should be provided about melanoma; for instance, authors correctly state that UV is the main risk factor, but they should also mention male sex, fair skin, amount of moles, and age (PMID: 34638427). Please expand this section.

Reply: In the Introduction more information on melanoma risk has been added as requested. The indicated reference has also been added.

  1. Line 42: “The incidence of melanoma is steadily increasing worldwide”; this statement requires to be justified by a reference.

Reply: The appropriate references have been added, as suggested.

  1. Line 61: “Taxanes are already successful in treating melanoma”; this statement requires to be justified by a reference.

Reply: The appropriate references have been added, as suggested.

  1. Line 61-65: this part is poorly written and sounds not scientific. Please rephrase it.

Reply: This part of the manuscript has been changed as suggested.

  1. Line 70: “It was also tested that betulinic acid has the ability to sensitize”; it is not clear to what betulinic acid is able to sensitize cells. Please rephrase and specify.

Reply: This paragraph has been extended and relevant references were added.

  1. Line 79 “contributes”.

Reply: The sentence has been changed as suggested.

  1. I do not understand why the measurements were performed at only 72h hours after treatment, and were not monitored over the time at 0, 24, 48 and 72h. Do authors have any data about previous timepoints?

Reply: In the "Materials and methods" section, the explanation has been added referring to the specific citations. All experiments were carried out during 72 hours of incubation. The 72h incubation time is the average doubling time for all melanoma cell lines tested.

  1. Figure 1: Authors must perform the same experiment on normal human keratinocyte cells HaCaT treated with BA. This is crucial to demonstrate that the reduction of cell viability observed in melanoma cells would not affect normal melanocytes.  

Reply: The MTT test was performed for normal keratinocytes, on which betulinic acid was tested. The Figure 1 and its description have been updated.

  1. A mechanistic insight would be highly recommended. Is BA administration able to increase intracellular ROS production?

Reply: To find out molecular mechanism(s) of action of betulinic acid on melanoma, the apoptosis test was determined using flow cytometry analysis. The mechanism of inducing apoptosis may be related to the cellular accumulation of ROS, which has been assessed in studies by other authors. The appropriate explanation and references were provided.

  1. Line 180: “Due to the growing number of cases of melanoma, new therapeutic methods are  sought. It is certainly related to the increased detection of this cancer, as well as the increasing exposure to ultraviolet (UVB) radiation”. These sentences are not connected each other; please rephrase.

Reply: The mentioned part of the manuscript has been changed.

  1. In the discussion section authors should mention also the other therapeutical approaches currently under evaluation for melanoma treatment, including coupling natural compounds and chemotherapeutics (PMID: 35453297). Indeed in line 216 they suggest coupling the BA with chemotherapeutics. This  point will strongly support the author’s design of study.

Reply: This paragraph was significantly enlarged by adding information on both the tested drug combinations and the current trends in searching for compounds that would lead to the accumulation of ROS in cancer cells. The indicated reference has also been added as suggested.

Reviewer 2 Report

In this manuscript, the authors examined the cytotoxic and anti-proliferative effects of betulinic acid, paclitaxel, and docetaxel in four melanoma cell lines. The authors found that all three drugs decreased the viability of melanoma cells. Furthermore, betulinic acid had no cytotoxic effect on normal cells, with the level of LDH significantly lower compared to melanoma cell lines. Finally the authors showed additive interactions when combining betulinic acid with paclitaxel or docetaxel. This is a well-written manuscript that sheds further light on the combination of betulinic acid and taxanes as a potential therapy for further investigation in melanoma. 

I have one minor query. Was the cytotoxicity of paclitaxel and docetaxel also investigated? 

Author Response

Thank You very much for Your time and Your review of our manuscript.
All comments indicated by the Reviewer were taken into account and the appropriate corrections were made to the manuscript.
We hope You will be satisfied with the corrections we have made.

"I have one minor query. Was the cytotoxicity of paclitaxel and docetaxel also investigated?"

Reply: The cytotoxicity of paclitaxel and docetaxel has not been investigated. Paclitaxel and docetaxel are registered drugs and used in clinical practice. The concentrations tested by us for paclitaxel 0.001-0.5 µM and for docetaxel 0.0005-0.08 µM are significantly lower than the maximum plasma concentrations (Cmax) in clinical practice, which are 4.27 µM and 5.47 µM for paclitaxel and docetaxel, respectively. Such an explanation, along with the relevant literature citations, has been added to the description of the MTT test results.

Round 2

Reviewer 1 Report

Manuscript is improved and can be accepted for publication.